# Multicomponent Aquatic Training (MAT) Program for People with Parkinson’s Disease: A Protocol for a Controlled Study

**DOI:** 10.3390/ijerph19031727

**Published:** 2022-02-02

**Authors:** Juliana Siega, Dielise Debona Iucksch, Vera Lucia Israel

**Affiliations:** 1Department of Physical Education, The Federal University of Paraná, Coronel Francisco H. dos Santos, 100, Curitiba 81531-980, Parana, Brazil; dielise@gmail.com (D.D.I.); veral.israel@gmail.com (V.L.I.); 2Department of Prevention and Rehabilitation in Physical Therapy, The Federal University of Paraná, Coração de Maria, 92, Curitiba 80210-132, Parana, Brazil

**Keywords:** Parkinson’s disease, physiotherapy, protocol, exercise, water exercise, aquatic skills, physical activity, quality of life, hydrotherapy

## Abstract

Introduction: The complications from Parkinson’s disease (PD) are directly related to decreased muscle function, balance deficits, and independence loss. Practicing aquatic exercises can minimize these symptoms and slow disease progress. Objective: To develop a Multicomponent Aquatic Training (MAT) protocol for people with PD between stages 1 and 4 of the Hoehn and Yahr scale. Methods: The protocol is for a single blind controlled clinical trial. The sample will comprise of people with PD between stages 1 and 4 in Hoehn and Yahr scale, divided into a control group and MAT group (who will participate in the MAT). Musculoskeletal function, functional mobility, and balance will be the primary outcomes of interest, assessed with an isokinetic dynamometer, the Five-Times-Sit-to-Stand test (FTSST), the Timed “Up and Go” test (TUG), the 6-m gait speed test, the Berg Balance Scale (BBS), and a force platform. Quality of life (QOL), activities of daily living (ADL), and motor aspects will be the secondary outcome measures, assessed with the Parkinson’s Disease Questionnaire (PDQ-39) and Unified Parkinson’s Disease Rating Scale (UPDRS), sections II and III. The MAT will be 12 weeks long, with two 50-min sessions per week. The outcome measures will be assessed before and after the interventions. Discussion: This study is expected to establish parameters to prescribe and monitor a MAT program for people with PD in stages 1 to 4 in the Hoehn and Yahr scale, respecting individual progress and assisting the professionals in their procedure with these people.

## 1. Introduction

Parkinson’s disease (PD) is chronic, neurodegenerative, and progressive [1]. It affects the midbrain substantia nigra, reducing the availability of dopamine and other neurotransmitters [1]. PD is the fastest-growing neurodegenerative disease in the world, as its prevalence increased from 2.5 million people in 1990 to 6.1 million in 2016 [2]. The prevalence rate in Brazil is estimated at 3.3% in people 65 years old or more, totaling at least 200,000 individuals who live with the disorder [1]. This poses a challenge to the public health system, with an increasing impact on Brazilian society [1].

PD causes motor and nonmotor impairments that interfere with the patient’s independence [3]. Its cardinal signs are bradykinesia, muscle stiffness, and tremor at rest [3]. The diagnosis requires that bradykinesia is associated with at least one of the other symptoms [3]. The Hoehn and Yahr Scale (HY) was developed to assess and classify people with PD according to the degree of incapacity and staging of the disease [4,5]. This scale has five stages verified according to the progression of the signs and symptoms, ranging from (1) unilateral disease to (5) bed- or wheelchair-ridden [4,5].

The treatment for PD is interdisciplinary [6]. Drug therapy is based on levodopa supplementation, a precursor medication of the neurotransmitter dopamine [7]. However, its continuous use has side effects and causes tolerance [7]. Physical therapy with exercises aims to preserve mobility and keep the muscles active to maintain the person’s independence and quality of life [6]. Another effect of physical exercise in PD is the increase in the brain neurotrophic factors and neurotransmitters and the decrease in neuronal degeneration in the initial stages of the disease [8]. Multicomponent physical exercise is a modality that, rather than isolated training, targets various aspects of physical function, such as cardiorespiratory fitness, muscle resistance, and balance [9,10]. It also promotes psychological and social well-being, corroborating the American College of Sports Medicine (ACSM) guidelines [9,10,11]. An additional possibility is to perform aquatic exercises (AE) [10].

AE is a resource widely used in the rehabilitation of physical function in people with PD, and has been highlighted as essential to the rehabilitation process [10]. The physical properties of water, such as resistance, hydrostatic pressure, and buoyancy, are different from those found on land [10]. Hence, AEs activate and improve motor control in a safe setting, with little risk of falls [10]. Its benefits include improved movement amplitude, balance, muscle strength, and flexibility [10]. They help control bradykinesia, optimizing overall body movements and, as a result, their activities and participation [10]. These assumptions are supported by two recent systematic reviews and meta-analyses [12,13], where it was shown that water-based exercise was more advantageous than land exercise for mobility and balance for people with PD [12,13].

Given all these characteristics, water may be a facilitating environment for people with greater motor difficulties [14,15,16]. However, most of the studies conducted in water with these populations approach the mild and moderate stages of the disease, classified between stages 1 and 3 in the HY scale [14,15,16].

The effects of aquatic programs on the more advanced stages (4 and 5), however, have not been a primary focus as it is difficult for these patients to join rehabilitation programs. Moreover, patients in stage 5 are no longer as independent as necessary to participate in a program without direct supervision. Therefore, we need strategies to attend to people in stages 1 to 4, aiming to delay the evolution of the disease and symptoms and progress into stage 5, as much as possible. Given the above, the objective of this study is to develop a Multicomponent Aquatic Training (MAT) protocol for PD patients in stages 1 to 4 on the HY scale. Sharing this protocol with the scientific community is the first step of a wider project objective to analyze the effects of MAT on the physical performance and cognitive state of people with PD.

## 2. Method

### 2.1. Study Design

This is a protocol for a single blind controlled clinical trial with two parallel groups, into which the participants will be allocated.

The MAT study protocol was developed based on the Standard Protocol Items: Recommendations for Interventional Trials (SPIRIT) [17]. The project was approved by the Ethics Committee of the Department of Health Sciences at the Federal University of Paraná, Curitiba, Paraná, Brazil (CAAE no. 66781417.4.0000.0102 and certification no. 2.200.372), and filed in the Brazilian Registry of Clinical Trials under RBR-6hnqcv. This is a Multicomponent Aquatic Training program, named MAT, to be conducted with participants with a clinical diagnosis of PD. The enrolled PD patients will be assessed at baseline (i.e., before dividing them into groups) and after three months. Any changes made in the protocol will be reported to the Research Ethics Committee and changed in the Brazilian Registry of Clinical Trials.

### 2.2. Study Setting

The study will be carried out in Curitiba, the capital of the state of Paraná, Brazil, where the Association of People with Parkinson’s Disease of Paraná (APPP) is located. The APPP is a nonprofit association founded 20 years ago, serving as a municipal, state, and federal public utility service that benefits about 1100 people with PD [18].

### 2.3. Participants

Individuals with PD residing in Curitiba or its metropolitan area will be invited to participate in the study, which will be advertised at APPP and in digital media.

The eligibility criteria for inclusion in the study are people of both sexes, diagnosed with idiopathic PD in stages 1 to 4 on the HY scale, who agree to participate in the research and sign the informed consent form (ICF), able to walk unassisted, and who have a medical certificate to perform physical exercises in a heated pool.

The exclusion criteria are diseases that cause vestibular or balance changes, or sensory, visual, or auditory deficits that hinder individuals from understanding the verbal or visual commands; absolute or relative contraindications to heated pool attendance; changes in the dosage or parameters of PD-related medication throughout the study as well as the implementation or withdrawal of Deep Brain Stimulation (DBS); change in physical activity levels during the study; being part of any other research up to three months before the first assessment of the study; not agreeing to or not signing the ICF; and withdrawing from the study.

All assessments and interventions will be carried out in the “on” period of medication for participants who meet the inclusion criteria.

### 2.4. Intervention Group

After being divided, the participants will be allocated into one of the two study groups: either the control group (CG) or the Multicomponent Aquatic Training program group (MAT group).

The project team will be comprised of physical therapists (experienced in both aquatic physical therapy and PD) and physical therapy students. The physical training will be offered free of charge to the participants, who will be asked not to start any other type of physical training during the study.

### 2.5. Multicomponent Aquatic Training (MAT)

The MAT was developed with PD patients classified between stages 1 to 4 in HY scale as the target population. The individuals allocated to the MAT group will be enrolled in a 12-week program with two 50-min sessions per week with at least 72 h between sessions, totaling 24 sessions.

The activities will take place in the pool at the Ana Carolina Moura Xavier Rehabilitation Hospital Center, in Curitiba, Paraná, Brazil. Due to the size and depth of the pool (10.75 m × 2.90 m; and 1.10 m deep), the groups will be limited to no more than six participants at a time. The pool will be kept at a mean temperature of 33 degrees Celsius.

Despite training being carried out in groups, there will be an experienced physical therapist and three physical therapy students in the pool to assist and instruct the patients while performing the exercises. Hence, individual attention will be provided, minimizing the risk of incidents in the water. There will also be another two physical therapy students outside the pool to provide support. These professionals will also record the arterial pressure (AP) and heart rate (HR) before and after the training, also being available in case any participant requires help during the intervention.

All training sessions will be separated as follows: functional mobility training and adjustment to the water (about 10 min); strength and muscle power exercises (about 10 min); balance or neuromotor exercises (about 20 min); and relaxation/cooldown (about 10 min). The MAT is described in detail in Annex 1, organized according to Israel’s five phases for learning aquatic motor skills [19], namely: (1) description of the activity, (2) main physical and thermal properties of the water involving the immersed body, (3) parameters of exercise progression, (4) transference of skills to land, and (5) classification of mobility in the ICF domains of activities and participation.

The training will be approached by progressively increasing volume and intensity to stimulate the participants’ neuroplastic potential [20]. Accordingly, every 4 weeks, training will be adjusted, with an increase in the degree of difficulty and complexity of the exercises. Table 1 describes the order and organization of the activities per week and intervention varieties, specifying which mobility/adjustment, muscle strength, balance, and relaxation training should be used.

### 2.6. Control Group

The participants allocated to the control group will be instructed not to change their usual activities throughout the study. Once the experimental period is completed, the intervention will be offered to the control participants free of charge.

### 2.7. Adverse Results

Adverse events will be monitored and prevented as the participants will be accompanied during their transition to the pool, throughout the exercise, and when their HR and AP are measured, always monitoring the intensity of their effort during the physical activity. During aquatic physical training, the participants will be individually accompanied by physical therapists and/or a support team inside and outside of the pool to ensure they do not perform the exercises incorrectly, thus avoiding aquatic accidents, such as momentary drownings.

In addition, the professionals will record the AP, HR, and breathing rate of participants before and after the training. In the event of any falls or incidents related to the subjects’ health status occurring during the intervention, the events will be recorded accordingly. During the training, the participants will also be monitored with the perceived exertion scale (BORG C20) to ensure they do not exceed the recommended intensity level. Participants must remain between 13 and 17, which would correspond to 66% and 80% of maximum voluntary strength production, respectively, as recommended by the American College of Sports Medicine 2014 [11]. If any musculoskeletal injuries occur, the training will be halted.

If any of the participants interrupt the protocol, follow-up data will be collected, and the adverse effects will be reported in the publication of the results. Delayed onset muscle soreness following physical training will not be considered an adverse effect as it is an expected result of the exercise. In the event of muscle soreness, participants will be referred for cryotherapy.

### 2.8. Recruitment

The participants will be informed about the project via a phone call and then invited to participate in the initial assessment. Before beginning the assessment, they will be informed of their voluntary participation in the study and sign the informed consent form—which has been revised and approved by the Ethics Committee of the Department of Health Sciences at the Federal University of Paraná. The study will be conducted in compliance with Brazilian Resolution 466/12, which is in line with the World Medical Association Declaration of Helsinki.

Other assessments planned for the pre-intervention period will be scheduled for the participants who meet the inclusion criteria and do not meet any of the exclusion criteria, as schematized in Figure 1.

The research team will read the informed consent form together with the participant and answer any questions they might have. Once the participants are fully informed and feel sure on their participation, they will sign two copies of the document (researchers and participants). All the collected data will be stored in secure locations, identified only with the participants’ code. The data will be used for publication in scientific journals without identifying any of the participants. In addition, at the end of the study, the participants will be handed a report with the data of their initial and final assessments. The data will be published regardless of the effects of the intervention—i.e., whether they are positive or negative.

### 2.9. Separation of the Groups and Allocation Concealment

The allocation of the 2 groups will be based on patient preference. Before completing the pre-intervention assessments, the participants will be divided into two equal-sized groups. The researcher responsible for the assessment will not know which group each participant is allocated to.

The participants will be instructed not to tell the researchers who assess them which study group they are a part of. Thus, the researchers who conduct the assessments will be blinded to the allocation and size of the groups.

### 2.10. Assessment

All assessments will be conducted before and after the intervention. A clinical assessment will be also conducted to classify the disease staging with the HY scale, which will be used only at the beginning due to its classificatory nature. Due to the amount of tests and measures, and to prevent participant fatigue, the assessments will be over 2 days.

### 2.11. Primary Outcome Measures

Musculoskeletal function, functional mobility, and balance were chosen as primary outcome measures because they are directly affected by the progression of the disease and are susceptible to interventions.

Muscle strength will be assessed with an isokinetic dynamometer (Biodex, System4Pro™, Shirley, NY, USA). The contraction type will be set to the concentric/concentric mode, the angle velocities selected will be 90°/s and 120°/s [21], and the protocol will include two repetitions throughout maximal range of motion for ankle plantarflexor and dorsiflexor muscles, knee and hip flexor and extensor muscles bilaterally, with a 2-min break in between repetitions [22].

The Five-Times-Sit-to-Stand functional test will be used to assess the strength and power of the lower limbs [23]. Participants will start the test in a seated position and will then be instructed to stand up and sit down five times on an armless chair (43 cm) as fast as possible, with the arms crossed on the chest [23]. The movement is repeated five times in a row without interruptions, being timed throughout the test [23]. The necessary equipment includes a fixed, armless, four-legged chair with back support [23]. If the person takes longer than 16 s to perform the task, they are categorized as being at a risk of falls [23].

Functional mobility will be further assessed with the Timed “Up and Go” Test (TUG), which consists of timing how long the person takes to stand up from a standard armchair (approximately 46 cm high), walk for 3 m, turn, walk back to the chair, and sit down again [24]. If they take longer than 16 s to complete the test, they are categorized as being at a risk of falls [23].

The 6-m walk test will be used to assess functional gait. It will be performed in a 10-m course, timing the walk on the 6 middle meters, excluding the acceleration and deceleration phases [23,25]. Three attempts will be collected at participants’ self-selected walking speeds; then the mean distance walked (18 m) will be divided by the sum of the three trial times. A speed slower than 0.98 m/s indicates a risk of falls for people with PD [23].

The Berg Balance Scale (BBS) will be used to assess stability and anticipatory postural control, as well as the association of strength, dynamic balance, and flexibility necessary for postural control [26]. The BBS has 14 items, each one scored from 0 to 4, thus totaling a maximum score of 56 points [26]. Higher values indicate better performance [26]. Conducting the BBS requires a chronometer, a ruler, one armless chair, and one armchair and it takes about 20 min to be performed [26,27]. Results lower than 47 points indicate a risk of falls for people with PD [23].

Postural control during quiet standing will be assessed with a force platform (model OR-06, AMTI, Watertown, MA, USA). The protocol requires the participant to stand still for 30 s, with arms at the side, for three times in each foot position, in the following order: (i) feet together with eyes open; (ii) feet together with eyes closed; (iii) semi-tandem feet with eyes open; (iv) semi-tandem feet with eyes closed.

The mean result of the three trials in each position will be used to assess postural control based on the sway of the center of pressure, calculated in the anteroposterior and mediolateral directions. The total length of the center of pressure displacement, anteroposterior and mediolateral amplitudes of the center of pressure, and the area of displacement of the center of pressure will be obtained from analyses made in MATLAB 9.1.0 (The MathWorks Inc., Natick, MA, USA).

### 2.12. Secondary Outcome Measures

Quality of life, motor function, and activities of daily living will be assessed as secondary outcome measures, as they are strongly influenced by the primary outcome measures.

Quality of life will be assessed with the Parkinson’s Disease Questionnaire (PDQ-39) [28,29]. The questionnaire has 39 questions divided into eight dimensions, namely: mobility, activities of daily living, emotional well-being, stigma (which assesses social difficulties in PD), social support, cognition, communication, and bodily discomfort [28,29]. The total score ranges from 0 to 100—the lower the score, the better the perception of health status [28,29].

Motor function and activities of daily living will be assessed with the Unified Parkinson’s Disease Rating Scale (UPDRS), sections II and III [30]. The UPDRS is an assessment scale for the patients’ signs, symptoms, and certain activities based on self-report and clinical observation [30]. It has 42 items distributed into four parts: mental activity, behavior, and mood (I); activities of daily living (II); motor examination (III); and drug therapy complications (IV) [30]. Each item scores from 0 (tending to normal) to 4 (greater impairment from the disease) [30].

Adherence and dropouts will also be collected to determine the tolerability and satisfaction of participants in the MAT.

### 2.13. Other Outcome Measures

The AP and HR will be measured before and after every assessment, and before and after every MAT session.

### 2.14. Sample Size

A sample size calculation was used to determine the number of participants required for the present study. The calculation was based on the prevalence of PD in Curitiba, Paraná. The sample calculation was performed using GPower 3.1 software, which stipulated a minimum sample of 30 individuals, assuming an effect size of 0.25 on a probability distribution F, whose value consisted of a mean distance between the sample mean and the population mean; a Type I error equal to 0.05 and analysis power equal to 0.84. Thus, the study will require 30 individuals with PD—15 allocated in the MAT group, and 15 in the control group. However, an extra 10% of participants will be added in order to avoid possible drop outs.

### 2.15. Statistical Analysis

The normality of data distribution will be assessed with the Shapiro–Wilk test and expressed as mean (±standard deviation), minimum, and maximum values. The variables analyzed will be compared between initial and final assessment and between groups with a mixed-model ANOVA. It is recommended that the effect size and statistical power be also published.

SPSS 20.0 statistical software will be used in all the analyses. A *p*-value < 0.05 will be used to determine significant differences.

## 3. Discussion

This study aims to develop a MAT program and determine its effects on muscle function, balance, and mobility in people with PD. The aquatic environment was chosen because it provides characteristics that can promote facilitation, support, and movement resistance. This is important for motor learning and re-learning in disabled populations.

Systematic reviews with meta-analyses and experimental studies highlight that aquatic exercises are a safe and feasible option for people with PD, as they provide various benefits related to motor, cognitive, psychological, and social aspects [8,16,31,32,33]. However, the prescription of aquatic exercises still depends on the definition of parameters—e.g., the exercises to be performed, the intensity of the training, the number of sets and repetitions, the duration of the intervention program, and weekly frequency. Moreover, there has been little study of the prescription of aquatic exercises for individuals with PD stage 4 on the HY scale, thus the potential benefit of aquatic exercises for this population is unknown [14].

Hence, this training protocol seeks to establish parameters for the prescription and monitoring of a MAT program for people with PD in stages 1 to 4 on the HY scale. To this end, the MAT proposes and respects individual and collective progress, either using water resistance or not. Therefore, it is expected that the combination of exercises for adjustment to the aquatic environment, mobility training, strength and power training, neuromotor exercises, and Ai-Chi exercises, will enhance the benefits of the individual therapies.

MAT was created based on the European Guidelines for PD [5], and the premises were to train functional mobility, lower limb muscle strength and power, and balance. All sessions were organized according to the same framework, and split into 3 phases: warm-up, main training, and cool down. The complexity and the intensity of the exercises will progress every 4 weeks. MAT follows neuroplasticity and motor learning principles, where it is recommended that a minimum period of 4 weeks is used before including new challenges to the activity [34].

The warm-up exercises are related to functional mobility and adaptation to the aquatic environment. Most include regular functional movements, such as walking in different directions, spinning, jumping, running, squatting, and diving to control breathing. Due to inherent trunk stiffness in those with PD, there will be axial exercises with spinning in all sessions.

The main training has exercises to develop lower limb muscle strength and power, as well as balance. Kicking and leg abduction/adduction were chosen as they activate large muscle groups, and also demand single foot support while the opposite limb combats hydrodynamic properties. Exercise progression was designed to be both quantity and load-based, respecting individual tolerance, as the participants will have different disability levels.

The balance training will include negotiating obstacles, changing postures, fluctuating postural control, using floating devices to increase instability during upright movements. The aquatic environment destabilizes the immersed body, stimulating postural control adjustments while reducing risks in case of falls [35]. The progression was based on the quality of movement and the decrease in upper limb support.

Finally, to relax and cool down, Ai-Chi sequences will be used. Respiratory control associated with big ranges of motion allows for simultaneous relaxation and stretching [36].

The aquatic environment is important for developing motor skills, and can be used as a resource for learning at different levels of physical disability. In populations such as PD, this learning should transfer to land. This would eventually increase the motor skillset when facing activities of daily living [35]. For this transfer of skillsets, Appendix A presents the expected conditions related to the main motor symptoms of PD, and also the International Classification of Functioning domains. The target is to extend the learning obtained from the aquatic environment through an increase in social participation of persons with PD. 

Lastly, this intervention protocol was developed to test a clinical intervention in this population, complying with the methodological recommendations necessary to obtain high quality results [23].

This clinical trial may have the following limitations: (1) difficulty recruiting individuals and having them adhere to a 12-week study plus the pre- and post-intervention assessment period; (2) the inability of some participants getting to the center where data collection will be made, especially those with stage four PD; (3) the MAT proposed may not be enough to cause favorable changes in all the stages of the disease, due to variability in participant characteristics and disease progress.

Furthermore, the protocol does not include: (1) the analysis of neuromuscular parameters (such as electromyography and reaction time) to understand the mechanisms involved in gaining muscle mass and strength; (2) the use of individual waterproof heart monitors during the intervention (aquatic heart rate meters, for instance) to follow up the heart rate oscillations. All these limitations must be considered in further research.

If the MAT provides positive results in terms of controlling the progress of PD symptoms with the parameters assessed, similar studies may be conducted to assist this population. Furthermore, as the protocol is described in detail, it can be used in public health and clinical practice, helping professionals and patients handle the symptoms and slow the progress of the disease.

## 4. Conclusions

This study expects to establish parameters to prescribe and monitor a MAT program for people with PD in stages 1 to 4 on the Hoehn and Yahr scale, respecting individual progress and assisting the professionals in their procedure with these people.

## Figures and Tables

**Figure 1 ijerph-19-01727-f001:**
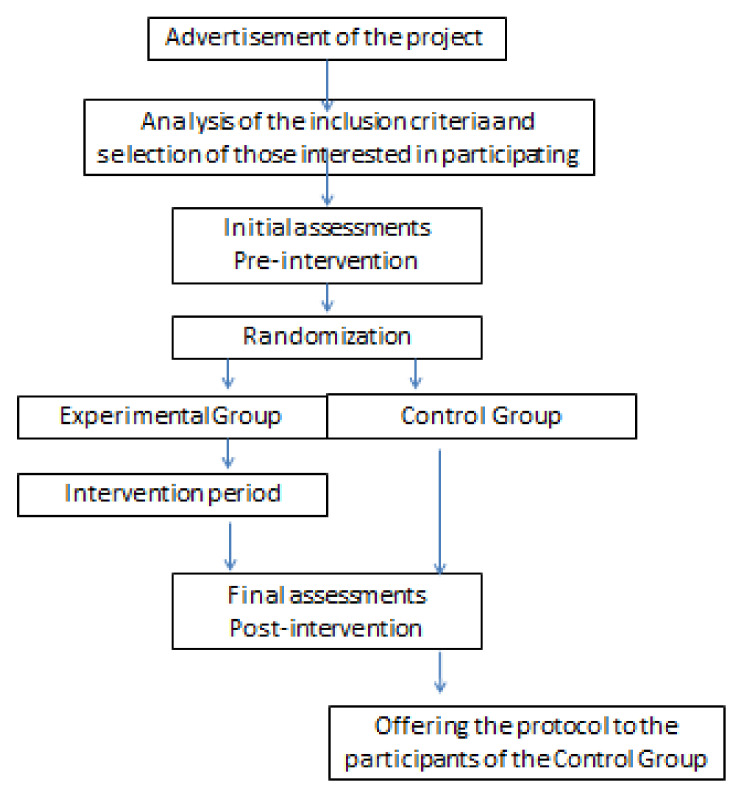
Study flowchart.

**Table 1 ijerph-19-01727-t001:** Weekly organization of the MAT.

Week	Phase 1 Gait and Adjustment Training	Phase 2 Strength Training	Phase 3 Balance Training	Phase 4 Relaxation
Weeks 1, 2, 3, 4	Training 1 (day 1) Training 2 (day 2)	Training 3	Training 6	Training 9
Weeks 5, 6, 7, 8	Training 1 (day 1) Training 2 (day 2)	Training 4	Training 7	Training 10
Weeks 9, 10, 11, 12	Training 1 (day 1) Training 2 (day 2)	Training 5	Training 8	Training 11

## Data Availability

As this paper describes a protocol, the main data is its own complete description as shown on Appendix A.

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
