# Peer review of "Multicomponent Aquatic Training (MAT) Program for People with Parkinson’s Disease: A Protocol for a Controlled Study"

_ijerph, 2022, doi:10.3390/ijerph19031727_

Round 1
Reviewer 1 Report
The present paper describes a protocol that aims to test the effect of water exercising on patients with stage 1 to 4 Parkinson’s disease (PD) according to Hoehn and Yahr scale. This subject is of great interest for the field as tools to slow down the disease progression are lacking and aquatic exercising has already proven some efficacy to improve patient’s condition. The paper precisely describes what are going to be these trainings, how these trainings are going to be organized, realized and supervised. The most important novelty of the paper is the focus on a wider population of PD patients as most of the previous studies were realized on patients with stage 1 to 3 and not stage 4. The manuscript appears globally well written and very detailed which is very appreciated. But a few minor points need to be addressed prior to publication.
Certain references (including 1 and 2) are not the original papers where the information was first reported but rather a paper that refer to the original paper. Authors should cite the original paper each time.
Section 2.3 The authors mention the exclusion of patients that have changes in their dosage of levoda, what about if changes occurs in other treatments used in coordination with levodopa such as dopamine agonists, COMT inhibitor, MAO inhibitor etc..? also, what about patients with DBS ?
Section 2.4 The authors should add information about how they are going to divide these patients into both the MATG and CG groups. These two groups should be composed of the most similar population of patients with PD regarding disease stage and treatment. It is even more important if the number of patients with stage 4 are usually low numbers as trying the MAT on patients with an advanced stage is also a particular novelty of the study. Also do the authors have a minimal number of patients with stage 4 to reach prior to start the study?
Line 124 The authors should specify how deep the pool is going to be. Are the patients going to be totally free of their movement?
Line 169 Globally regarding the protocol, no stretching seems included, just curious whether there is a specific reason for that?
Annexe1 phase2: what does “prescription” mean here? Maybe another word would fit better. Also the table needs to be adjusted so we can read what is vertically written at the left side of the table.
Author Response
Curitiba, January 2022
Dear reviewer,
We thank you for your comments and recommended revisions for our manuscript. We have adapted the manuscript according to your suggestions and feel that it has greatly improved the quality of our work.
We emphasize that we also sent for English editing.
Reviewer 1
The present paper describes a protocol that aims to test the effect of water exercising on patients with stage 1 to 4 Parkinson’s disease (PD) according to Hoehn and Yahr scale. This subject is of great interest for the field as tools to slow down the disease progression are lacking and aquatic exercising has already proven some efficacy to improve patient’s condition. The paper precisely describes what are going to be these trainings, how these trainings are going to be organized, realized and supervised. The most important novelty of the paper is the focus on a wider population of PD patients as most of the previous studies were realized on patients with stage 1 to 3 and not stage 4. The manuscript appears globally well written and very detailed which is very appreciated. But a few minor points need to be addressed prior to publication.
Thank you for the review and comments that will make our text even better.
Certain references (including 1 and 2) are not the original papers where the information was first reported but rather a paper that refer to the original paper. Authors should cite the original paper each time.
Response: We replaced the citations with the original articles.
Section 2.3 The authors mention the exclusion of patients that have changes in their dosage of levodopa, what about if changes occurs in other treatments used in coordination with levodopa such as dopamine agonists, COMT inhibitor, MAO inhibitor etc..? also, what about patients with DBS ?
Response: Thanks for the observation. Not only will levodopa will be monitored, but all other PD-related medication. Changed in text. In relation to DBS, we also considered as exclusion factor the installation or removal during the training period.
Section 2.4 The authors should add information about how they are going to divide these patients into both the MATG and CG groups. These two groups should be composed of the most similar population of patients with PD regarding disease stage and treatment. It is even more important if the number of patients with stage 4 are usually low numbers as trying the MAT on patients with an advanced stage is also a particular novelty of the study. Also do the authors have a minimal number of patients with stage 4 to reach prior to start the study?
Response: This is actually a great question. Our research group has a partnership with Parana’s Parkinson Association, which has approximately 1100 members with PD. Unfortunately, the Association does not have updated figures about the HY incidence. To better represent the HY 4 in our sample, we also consulted the Brazilian Ministry of Health epidemiological database but couldn’t find reliable info. This way we ended up using a study about prevalence of PD in Germany as baseline. This study (DOI: 10.1159/000477165) mentions the proportion of HY 1 and 2 at 43%; HY 3 at 35% and HY 4 at 17%. Therefore, we intend to have around 3 participants on each group with HY 4.
Line 124 The authors should specify how deep the pool is going to be. Are the patients going to be totally free of their movement?
Response: The depth of the pool is 1.10 meters, being approximately at the height of the xiphoid process. Small changes will not be relevant as most exercises are for lower limbs. When performing exercises for the upper limbs, patients will be instructed to keep their shoulders at the height of the water surface. We included in the text the depth of the pool on item 2.5.
Line 169 Globally regarding the protocol, no stretching seems included, just curious whether there is a specific reason for that?
Response: Not specifically, but within the Ai-Chi method, stretches are covered in cool down/relax. Upper limbs, lower limbs and breathing movements are more complex and active movements this way.
Annexe1 phase2: what does “prescription” mean here? Maybe another word would fit better. Also the table needs to be adjusted so we can read what is vertically written at the left side of the table.
Response: We use the term "prescription" in the definition of the word, in the sense of respecting the order, time and volume of this moment of training, which refers to strength and power training.

Reviewer 2 Report
- Consider adding adherence and tolerability components. What good is a program that benefits the individual from a musculoskeletal perspective, but that they don't enjoy or don't show up to? These should be secondary outcomes that are reported transparently. As far as tolerability goes, I would strongly recommend finding a survey or doing an exit interview to address how the participants honestly felt about the program itself.
- Table 1 is very confusing without reference to Appendix 1 in the caption.
- Will a physical activity questionnaire be given to the control group to record how much exercise they may have done during the intervention? I know group allocation is based on their preference, but people can change their minds. How will you control for compensatory rivalry?
- Line 163 is the first time you mention RPE -- maybe bring this up when talking about the training program, if they're supposed to stay within a given intensity for a given training session?
- Line 310 -- Should this be a limitation, or an exclusion criteria? It seems to me that if an individual is completely unable to make it to the center, either accommodations need to be planned for, or these individuals need to be screened out on the front end.
- Appendix 1: No 1 Phase 1, No 2 Phase 1, No 8 phase 3, No 9 phase 4, No 10 phase 4, and No 11 phase 4 do not have a prescription.
Author Response
Curitiba, January 2022
Dear reviewer,
We thank you for your comments and recommended revisions for our manuscript. We have adapted the manuscript according to your suggestions and feel that it has greatly improved the quality of our work.
We emphasize that we also sent for English editing.
Reviewer 2
- Consider adding adherence and tolerability components. What good is a program that benefits the individual from a musculoskeletal perspective, but that they don't enjoy or don't show up to? These should be secondary outcomes that are reported transparently. As far as tolerability goes, I would strongly recommend finding a survey or doing an exit interview to address how the participants honestly felt about the program itself.
Response: The observation is pertinent. At the end of the research, the characterization of the sample will bring aspects related to adherence and sample losses during the intervention program. The phrase "Adherence and dropouts will also be collected to determine the tolerability and satisfaction of participants in the MAT” was included at the end of the secondary results.
- Table 1 is very confusing without reference to Appendix 1 in the caption.
Response: Table 1 was described in Appendix 1, in the upper column with the "No" of training, immediately before the column "Number and type of training". We changed the nomenclature to "Training number" in order to make it clearer.
- Will a physical activity questionnaire be given to the control group to record how much exercise they may have done during the intervention? I know group allocation is based on their preference, but people can change their minds. How will you control for compensatory rivalry?
Response: During the initial assessments, participants will be instructed to maintain their usual physical activities, not starting any new physical activity during this period. These activities will be documented. As the character of PD is progressive and the intervention in a pool is complementary to the therapies on the ground, we will verify the additional effects/benefits of the MAT on the participants. If they start any new activity during this period, they will be excluded from the analysis to avoid interference.
About compensatory rivalry we will check every session for the attendance and based in your comments we’ll include an individual feedback session for each participant where their performance will be analyzed and discussed. Henceforth we do believe that the exercises progression will be a way to keep their motivation.
- Line 163 is the first time you mention RPE -- maybe bring this up when talking about the training program, if they're supposed to stay within a given intensity for a given training session?
Response: Thanks for the comment. We added the sentence "Participants must remain between 13 and 17, which would correspond to 66% to 80% of the voluntary production of maximum strength, respectively, as recommended by the American College of Sports Medicine 2014" after the information that they will be monitored by BORG C20.
- Line 310 -- Should this be a limitation, or an exclusion criteria? It seems to me that if an individual is completely unable to make it to the center, either accommodations need to be planned for, or these individuals need to be screened out on the front end.
Response: In this limitation, we referred more to practical issues such as the individual being assisted by a family member, if necessary. The location in which the MAT will be applied is a large city, with many patients coming from the metropolitan region. Even though there are adapted buses that pick up patients at home, there may be adverse conditions in which the patient cannot go. Regarding accessibility, the intervention will be applied in a hospital with complete accessibility, including access ramps and a lift in the pool.
- Appendix 1: No 1 Phase 1, No 2 Phase 1, No 8 phase 3, No 9 phase 4, No 10 phase 4, and No 11 phase 4 do not have a prescription.
Response: We are grateful for the observation, however the term "prescription" was only used in the strength training stage.

Reviewer 3 Report
1- General comment
Dear authors, my congratulations for the pertinent theme and for developing a study protocol before conducting the study. This phase is crucial because it will allow you to prevent critical errors during the intervention and improve the scientific robustness of your study. In general, the study protocol is well designed, but there is space for improvement. Thus, I recommend major revisions to improve the quality of the study protocol. Below, I provide some suggestions.
2- Abstract
Please, remove the headings, as suggested by the IJERPH norms.
LL15: Maybe instead of "experimental group", you can write "MAT group".
LL16: What do you mean by main results? Primary outcomes? Please, specify.
LL19: Since you wrote secondary outcomes, I suggest writing primary outcomes in LL16.
LL21: Please, change "sessions a week" to "sessions per week".
LL21: Please, change "All the results" to "The results".
LL22-25: In my opinion, the main aim of your study (not the protocol) is to analyze the MAT effects on physical performance and cognitive state (e.g., quality of life) in people with PD. Establishing parameters to prescribe and monitor the MAT program in people with PD will result from your results. Therefore, if effective (I hope so), you will be able to provide practical recommendations for researchers and coaches regarding training prescription in people with PD.
LL26: Please, consider including more relevant keywords (you can insert 3 to 10) to increase the visibility of your study protocol.
3- Introduction
The study's rationale is presented (well-structured introduction), and the research problem is defined. Nevertheless, I suggest that the authors add the project's main aims (e.g., 1- analyze the effects of MAT on physical performance; 2- analyze the effects of MAT on the cognitive state).
4- Methods
LL98-108: If possible, insert a table with the inclusion/exclusion criteria information. It will facilitate its reading.
LL99: Is the Hoehn and Yahr scale valid and reliable to diagnose the different PD stages? If possible, cite a study where the validity and reliability of the scale were assessed.
LL109: Who will conduct the initial screenings? Please, specify.
LL122: What will be the difference in days between sessions? 24 hours? 48 hours? Please, specify.
LL142: I guess that you want to say “progressively increasing volume (i.e., sets and repetitions) and intensity (in your case, it is more indicated the level of effort)” instead of “progressively load and intensity”. Please, rectify.
LL143: What do you mean by “the training is enhanced”? I suggest removing this sentence.
LL193: Why will you not randomly allocate the participants to each group? This procedure will help strengthen the internal validity of your study and prevent selection bias. Please, comment.
LL217: You can also use validated equations to estimate the absolute and relative power generated in the Five-Times-Sit-to-Stand test. Please, see the study developed by Alcazar et al. (https://doi.org/10.1016/j.exger.2018.08.006).
LL272: The sample size calculation lacks detailed information. How did you obtain a required sample size of 60 participants? Please, specify the calculation for better comprehension. If possible, define the statistical power (usually 80%) and the effect size according to previous literature (e.g., a meta-analysis). The GPower free software might be an option to estimate the sample size.
LL279: By mentioning parametric tests (i.e., ANOVA), you expect that your variables follow a normal distribution. However, if the assumption is violated, you will have to use non-parametric tests. Therefore, I suggest you add this information and specify what tests you will use in that case.
5- Discussion
LL288: Please, change Multicomponent Aquatic Training to MAT. When the abbreviation is defined first, you should use it throughout the text.
LL288-289: Here, you presented, in my opinion, the main aim of your study. I suggest that you do the same in the abstract and introduction section.
LL297: Please, change “are not much known” to “are poorly known”.
6- Conclusion
As mentioned before, I suggest you indicate that this study aims to analyze the effects of MAT on the physical performance and cognitive state of people with PD.
Author Response
Curitiba, January 2022
Dear reviewer,
We thank you for your comments and recommended revisions for our manuscript. We have adapted the manuscript according to your suggestions and feel that it has greatly improved the quality of our work.
We emphasize that we also sent for English editing.
Reviewer 3
2- Abstract
Please, remove the headings, as suggested by the IJERPH norms.
Response: The header was an edit made by the Journal before sending it to reviewers.
LL15: Maybe instead of "experimental group", you can write "MAT group".
Response: We appreciated the suggestion and changed the term "experimental group" to "MAT group".
LL16: What do you mean by main results? Primary outcomes? Please, specify.
Response: We changed the term "outcomes" to "results" and "main" to "primary". It was just out of the box. Thanks for highlighting this error.
LL19: Since you wrote secondary outcomes, I suggest writing primary outcomes in LL16.
Response: We changed the term "outcomes" to "results" and "main" to "primary". It was just out of the box. Thanks for highlighting this error.
LL21: Please, change "sessions a week" to "sessions per week".
Response: We change as suggested. Thank you.
LL21: Please, change "All the results" to "The results".
Response: We change as suggested. Thank you.
LL22-25: In my opinion, the main aim of your study (not the protocol) is to analyze the MAT effects on physical performance and cognitive state (e.g., quality of life) in people with PD. Establishing parameters to prescribe and monitor the MAT program in people with PD will result from your results. Therefore, if effective (I hope so), you will be able to provide practical recommendations for researchers and coaches regarding training prescription in people with PD.
Response: We partly agree with you. This protocol is part of a wider research project to be published in the near future. The objective you’ve mentioned aligns with this research project. However, in this article, our target was actually to describe the protocol that we will use as detailed as possible. The aquatic area lacks greater systematization of knowledge and therefore writing the protocol in detail would be relevant. Throughout the text, according to your requests, we were adding information and highlighting the changes.
LL26: Please, consider including more relevant keywords (you can insert 3 to 10) to increase the visibility of your study protocol.
Response: Excellent observation. We added the following keywords: aquatic skills, water exercise, physical activity, quality of life, Hydrotherapy
3- Introduction
The study's rationale is presented (well-structured introduction), and the research problem is defined. Nevertheless, I suggest that the authors add the project's main aims (e.g., 1- analyze the effects of MAT on physical performance; 2- analyze the effects of MAT on the cognitive state).
Response: As mentioned above, we believe that these are the goals to be achieved after the application of this protocol. For now, the results, discussion and conclusion shall be related to the current study. We still don’t have data to describe the effects of this exercise program. Anyway, we included the required info at the introduction section.
4- Methods
LL98-108: If possible, insert a table with the inclusion/exclusion criteria information. It will facilitate its reading.
Response: We appreciate the suggestion; however we chose to leave this information in text.
LL99: Is the Hoehn and Yahr scale valid and reliable to diagnose the different PD stages? If possible, cite a study where the validity and reliability of the scale were assessed.
Response: Yes, the Hoehn and Yahr Scale is already cited and referenced, it is reference 4 (MM Hoehn, MD Yahr.. Parkinsonism: onset, progression, and mortality. Neurology. 17.5 (1967) 427–442. doi: https ://doi.org/10.1212/WNL.17.5.427).
LL109: Who will conduct the initial screenings? Please, specify.
Response: The initial screenings will be conducted by 1 physiotherapist of our research group, who is not involved with this project.
LL122: What will be the difference in days between sessions? 24 hours? 48 hours? Please, specify.
Response: Sessions must be held at least 72 hours apart. We add this information to the text.
LL142: I guess that you want to say “progressively increasing volume (i.e., sets and repetitions) and intensity (in your case, it is more indicated the level of effort)” instead of “progressively load and intensity”. Please, rectify.
Response: Thank you, we corrected as suggested.
LL143: What do you mean by “the training is enhanced”? I suggest removing this sentence.
Response: We completed the sentence to clarify the idea of "enhanced". In fact, training progresses every 4 weeks.
"Hence, every 4 weeks, the training is enhanced, with an increase in the degree of difficulty and complexity of the exercises."
LL193: Why will you not randomly allocate the participants to each group? This procedure will help strengthen the internal validity of your study and prevent selection bias. Please, comment.
Response: The allocation into groups will be based on patients-preference. We understand that randomization is an important process to avoid bias and we are aware that it is a limitation to this study. On the other hand, we also consider that exposing participant to protocols to which they are not motivated can also be a source of bias and drop out.
LL217: You can also use validated equations to estimate the absolute and relative power generated in the Five-Times-Sit-to-Stand test. Please, see the study developed by Alcazar et al. (https://doi.org/10.1016/j.exger.2018.08.006).
Response: Thanks for the suggestion, we didn't know about this article. In the analysis of the pre and post moments and in the article that will effectively verify the effects of MAT, we will verify the possibility of using it. Thank you.
LL272: The sample size calculation lacks detailed information. How did you obtain a required sample size of 60 participants? Please, specify the calculation for better comprehension. If possible, define the statistical power (usually 80%) and the effect size according to previous literature (e.g., a meta-analysis). The GPower free software might be an option to estimate the sample size.
Response: Thanks for the observation. We revised the calculations and our sample should be 30. We changed the text to: “The sample calculation was performed by GPower 3.1 software,12 which stipulated a minimum sample of 30 individuals, assuming an effect size of 0.25 on a probability distribution F, whose value consists of a mean distance between the sample mean and the population mean; Type I error equivalent to 0.05 and analysis power equal to 0.84”.
LL279: By mentioning parametric tests (i.e., ANOVA), you expect that your variables follow a normal distribution. However, if the assumption is violated, you will have to use non-parametric tests. Therefore, I suggest you add this information and specify what tests you will use in that case.
Response: In the case of non-parametric data, instead of mean and standard deviation it will be median and interquartile range of 25% and 75%. For intergroup comparison - between groups - Wilcoxon will be used, and for comparison intragroup (within groups) Mann-Whitney test.
5- Discussion
LL288: Please, change Multicomponent Aquatic Training to MAT. When the abbreviation is defined first, you should use it throughout the text.
Response: We changed "Multicomponent Aquatic Training" to "MAT" throughout the text.
LL288-289: Here, you presented, in my opinion, the main aim of your study. I suggest that you do the same in the abstract and introduction section.
Response: Actually, the purpose of this article is the development of the protocol. To improve understanding and go to the meeting of your suggestion, we add this sentence at the end of the introduction: “Share this protocol with scientific community is the first step of a wider project objective which is to analyze the effects of MAT on the physical performance and cognitive state of people with PD.”
LL297: Please, change “are not much known” to “are poorly known”.
Response: We changed as suggested. Thank you.
6- Conclusion
As mentioned before, I suggest you indicate that this study aims to analyze the effects of MAT on the physical performance and cognitive state of people with PD.
Response: Perhaps during the first review the objective was not clear. Based on your and other reviewers' comments, we made a series of changes to clarify the objective: the development of the protocol. We hope it was enough. Thanks for all contributions.

Reviewer 4 Report
The study aimed to develop a Multicomponent Aquatic Training program and verify its effects on muscle function, balance, and mobility in people with PD.
When the text is examined, the research is written in the future tense. This shows that the study was not done.The study is presented in the form of a research proposal.
In addition, One of the most important parts of a study is the result and discussion section. There are no results in this study and the results of the study are not discussed. And its results are not explained with reasons by current researches.
There is no results section in the study. The discussion section is very short. After the study is completed, it can contribute to the field. However, it is my decision that this work which is incomplete, should not be published.
Author Response
Curitiba, January 2022
Reviewer 4
Dear reviewer,
We believe that the type and the objective of the present study was not understood. We are proposing here a Study Protocol, as specified on the Header of the submission page.
A protocol study aims to develop an intervention or training aquatic program for a specific population. We have included guidelines for recruitment, selection, assessment measures, and outcome analysis at the time this protocol is reproduced, but the greatest results of this study are the detailed descriptions of each exercise, which is presented on Appendix 1. We understand your point of view but this study is not incomplete, it is just structured differently of a randomized clinical trial comparing pre- and post-intervention.
Anyway, based on your comments and the others reviewers we added information related to objective and completed the discussion section to add clarity.
We appreciate your contributions.

Round 2
Reviewer 3 Report
Dear authors,
I read the revised manuscript, and I want to congratulate you on the excellent work in revising the manuscript as requested. In my opinion, the manuscript is ready for publication. Good luck with the experimental studies!
Best regards.
Reviewer 4 Report
Dear Authors, I supposed that your article was original article. Your reearch design is study Protocol.. I saw that and also İ am so sorry for this. You improved the your study. The study is ready for publication